# Antagonization of OX_1_ Receptor Potentiates CB_2_ Receptor Function in Microglia from APP_Sw/Ind_ Mice Model

**DOI:** 10.3390/ijms232112801

**Published:** 2022-10-24

**Authors:** Iu Raïch, Joan Biel Rebassa, Jaume Lillo, Arnau Cordomi, Rafael Rivas-Santisteban, Alejandro Lillo, Irene Reyes-Resina, Rafael Franco, Gemma Navarro

**Affiliations:** 1Molecular Neuropharmacology Laboratory, Department of Biochemistry and Physiology, School of Pharmacy and Food Science, Universitat de Barcelona, 08007 Barcelona, Spain; 2CiberNed, Network Center for Neurodegenerative Diseases, National Spanish Health Institute Carlos III, 28029 Madrid, Spain; 3Neurosciences Institut, University of Barcelona (NeuroUB), 08028 Barcelona, Spain; 4Bioinformatics, ESCI-UPF, 08003 Barcelona, Spain; 5School of Chemistry, Universitat de Barcelona, 08007 Barcelona, Spain

**Keywords:** orexin, cannabinoids, Alzheimer’s disease, activated microglia

## Abstract

Microdialysis assays demonstrated a possible role of orexin in the regulation of amyloid beta peptide (Aß) levels in the hippocampal interstitial fluid in the APP transgenic model. CB_2_R is overexpressed in activated microglia, showing a neuroprotective effect. These two receptors may interact, forming CB_2_-OX_1_-Hets and becoming a new target to combat Alzheimer’s disease. Aims: Demonstrate the potential role of CB_2_-OX_1_-Hets expression and function in microglia from animal models of Alzheimer’s disease. Receptor heteromer expression was detected by immunocytochemistry, bioluminescence resonance energy transfer (BRET) and proximity ligation assay (PLA) in transfected HEK-293T cells and microglia primary cultures. Quantitation of signal transduction events in a heterologous system and in microglia cells was performed using the AlphaScreen^®^ SureFire^®^ kit, western blot, the GCaMP6 calcium sensor and the Lance Ultra cAMP kit (PerkinElmer). The formation of CB_2_-OX_1_ receptor complexes in transfected HEK-293T cells has been demonstrated. The tetrameric complex is constituted by one CB_2_R homodimer, one OX_1_R homodimer and two G proteins, a G_i_ and a G_q_. The use of TAT interfering peptides showed that the CB_2_-OX_1_ receptor complex interface is TM4-TM5. At the functional level it has been observed that the OX_1_R antagonist, SB334867, potentiates the action induced by CB_2_R agonist JWH133. This effect is observed in transfected HEK-293T cells and microglia, and it is stronger in the Alzheimer’s disease (AD) animal model APPSw/Ind where the expression of the complex assessed by the proximity ligation assay indicates an increase in the number of complexes compared to resting microglia. The CB_2_-OX_1_ receptor complex is overexpressed in microglia from AD animal models where OX_1_R antagonists potentiate the neuroprotective actions of CB_2_R activation. Taken together, these results point to OX_1_R antagonists as drugs with therapeutic potential to combat AD. Data access statement: Raw data will be provided by the corresponding author upon reasonable requirement.

## 1. Introduction

Orexin (hypocretin) receptor-mediated signaling has been mainly studied from the point of view of endocrinology and in relationship to eating disorders. However, orexin-actions are also important from the point of view of neurodegenerative diseases. Regarding Alzheimer’s disease (AD), microdialysis assays showed that orexin treatment resulted in the regulation of amyloid beta peptide (Aß) levels in the hippocampal interstitial fluid of a transgenic model that overexpresses a mutant form of the amyloid precursor protein (APP). The effect of orexin was mediated by receptors as it was blocked by orexin receptor antagonists [1]. A further link between the orexigenic actions and the pathophysiology of AD comes from the role of orexin receptors in sleep-wake cycles, which are altered in AD patients (see [2] for review). In addition, a case-control study showed that (i) the levels of orexin in the cerebrospinal fluid (CSF) were significantly higher in AD patients than in non-demented controls, and (ii) a positive correlation of orexin and tau levels in the CSF of AD patients [3]. Furthermore, it has been observed that in the APP/PS1 transgenic AD model, orexin-A aggravated cognitive deficits by a mechanism that, at least in part, involved alterations in mitochondrial function [4]. A detailed review of the relationships between the orexigenic pathways and their potential impact on pathophysiological aspects of AD is found elsewhere [5].

Two main orexin neuropeptides have been discovered, orexin-A (HCRT1, O43612, orexin-A) and B (HCTR2, orexin-B), which arise from the precursor gene prepro-orexin and act via cell surface G-protein coupled receptors (GPCRs). Orexin-A is a 33 amino acid peptide with two intrachain disulfide bonds while OXB is a linear 28 amino acid peptide [6]. Orexins have neuroprotective and immunoregulatory properties (Couvineau et al., 2019) [7]. There are two types of orexin receptors, OX_1_R and OX_2_R, which belong to the class A family and are mainly coupled to heterotrimeric Gq proteins. Thus, they can activate phospholipase A2, C and D, diacilglycerol lipase and calcium ion-mediated responses [8,9]. The amino acid sequence identity of human orexin receptors is 64% [10]. Orexin-A shows approximately equal affinity for both receptors while OXB shows a higher affinity for OX_2_R than OX_1_R [11,12,13]. The mRNAs encoding the two receptors are both enriched in the brain and moderately abundant in the hypothalamus but display remarkably different distributions. OX_1_R mRNA is expressed in many brain regions including the prefrontal and infralimbic cortex, hippocampus, paraventricular thalamic nucleus, ventromedial hypothalamic nucleus, dorsal raphe nucleus, and locus coeruleus. OX_2_R mRNA is prominently expressed in complementary distribution including the cerebral cortex, septal nuclei, hippocampus, medial thalamic groups, raphe nuclei, and many hypothalamic nuclei including the tuberomammillary nucleus, dorsomedial nucleus, paraventricular nucleus, and ventral premammillary nucleus [14]. The orexigenic system is involved in the regulation of endocrine, autonomic, and behavioral responses to maintain homeostasis [15], being involved in processes such as motivation, sleep-wake, learning and memory [7]. It has been demonstrated that orexigenic function degenerates with age, and thus dysregulation can result in several cognitive and behavioral deficits [13].

The most abundant cell surface GPCR in the mammalian brain is the cannabinoid CB_1_ receptor (CB_1_R), which is expressed in both neurons and glia. A second cannabinoid receptor, CB_2_R, is less present in neurons but is expressed in glia. In particular, it is overexpressed in activated glial cells, for instance in activated microglia [16]. The canonical G protein to which both receptors couple is G_i_, i.e., the activation of cannabinoid receptors leads to decreases in the level of cytosolic cAMP and inactivation of protein kinase A-dependent pathways [6]. It has been previously shown in the cortex of AD patients that the expression of the CB_1_R is decreased and that this decrease correlates with hypophagia. In contrast, the CB_2_ receptor (CB_2_R) was upregulated and the expression increase correlated with higher glial marker expression and, importantly, with senile plaque score and Aß levels [17]. The therapeutic potential of targeting the CB_2_R has gained interest due to its involvement in microglial activation, which is one of the features of AD. Preclinical research has demonstrated in a variety of models that cannabinoids devoid of psychotropic effects and mainly targeting the CB_2_Rs have benefits in improving cognition while reducing neuroinflammation and preventing abnormal APP and tau processing (see [18,19] for review). 

The first aim of this study was to find out if the CB_2_R and the OX_1_R can establish direct receptor-receptor interactions. A second aim was to discover the interrelationships between the signaling mediated by these two receptors and their potential as therapeutic targets to combat AD, with special emphasis on evaluating their role in activated microglia.

## 2. Results

### 2.1. Direct Interaction of Cannabinoid CB_2_R and Orexin OX_1_R Receptors in a Heterologous Expression System 

We first aimed to assess whether the cannabinoid CB_2_ (CB_2_R) and orexin OX_1_ (OX_1_R) receptors might interact. Immunocytochemistry assays were performed in HEK-293T cells expressing the CB_2_R fused to YFP, the OX_1_R fused to Rluc, or both (Figure 1A). CB_2_R-YFP was detected by the YFP green fluorescence and OX_1_R-Rluc was detected by a mouse monoclonal anti-Rluc antibody and a secondary Cy3 conjugated anti-mouse IgG antibody. Results in Figure 1A show that when independently transfected, both receptors are expressed at the plasma membrane level and also at the cytosol. Moreover, in cells expressing both fusion proteins, receptors colocalize at the plasma membrane and intracellularly (Figure 1A). As colocalization does not demonstrate direct interaction, bioluminescence resonance energy transfer (BRET) assays were carried out in HEK-293T cells expressing a constant amount of OX_1_R-Rluc and increasing amounts of CB_2_R-YFP. A hyperbolic BRET saturation curve indicated a specific interaction between the receptors and the formation of CB_2_R-OX_1_R heteromers with the following parameters: BRET_max_ = 127 ± 7 mBU and BRET_50_ = 110 ± 20 mBU (Figure 1B). BRET_max_ results from donor/acceptor proximity and the number of CB_2_R-OX_1_R heteromers while BRET_50_ gives an idea of the affinity of the interaction. Accordingly, these parameters indicate that the interaction of the two receptors is strong. As negative control, GABA_B_-Rluc was used instead of OX_1_R-Rluc, and a linear signal was obtained, indicating the lack of interaction between CB_2_R and GABA_B_ receptors (Figure 1C).

### 2.2. OX_1_R Antagonists Potentiate CB_2_R-Mediated Signalling in Transfected HEK-293T Cells

Signaling studies were first performed in single transfected cells. CB_2_R couples to G_i_ protein, thus leading to the inhibition of adenylate cyclase and the decrease of intracellular cAMP levels. The concentration of this second messenger was measured in cells expressing CB_2_Rs and pretreated with forskolin to activate adenylate cyclase and increase the cAMP levels. In CB_2_R expressing cells, the selective agonist JWH133 induced a 70% decrease in forskolin-induced cAMP levels (Figure 2A). This effect was specific as it was completely blocked by pretreatment with the selective CB_2_R antagonist, SR144528. OX_1_R can couple to different G proteins depending on the heteromeric complexes in which the receptor is involved, hence we first performed assays assuming that it could couple to the G_i_ protein. Determination of cAMP levels in HEK-293T cells expressing the OX_1_R showed that orexin-A induced a 70% decrease of forskolin-induced cAMP levels and that SB334867, a selective antagonist, completely blocked this effect (Figure 2B). When similar assays were performed in cells coexpressing both CB_2_R and OX_1_R, the agonists of the two receptors exerted a significant effect that was reverted by the corresponding antagonists (Figure 2E). However, the CB_2_R antagonist not only blocked the JWH133-induced effect but also the effect of the OX_1_R agonist. This phenomenon is known as cross-antagonism and can be used as a print to detect CB_2_R-OX_1_R heteromers in homologous systems. In contrast, the OX_1_R antagonist blocked the effect triggered by orexin-A, but potentiated the effect of the CB_2_R agonist (Figure 2E). This result is relevant, as it suggests that OX_1_R antagonists might potentiate the neuroprotective effects mediated by CB_2_R. On the other hand, when activating the same cells with both agonists, orexin-A and JWH133, a similar effect was observed to that induced by the activation of only one of the receptors (Figure 2E).

Due to the capability of OX_1_R to couple to the G_q_ protein and subsequently increase the cytosolic concentration of calcium ion (Ca^2+^), assays were performed to measure this second messenger in cells expressing OX_1_R or both receptors. First, experiments were performed in cells expressing the CB_2_R and the calcium sensor GCaMP6; JWH133 stimulation did not lead to any effect, thus confirming that CB_2_R is not coupled to G_q_ (Figure 2C). Orexin-A in cells expressing the OX_1_ receptor or OX_1_ and CB_2_ receptor produced a transient increase of Ca^2+^ concentration that was blocked by the OX_1_R antagonist (Figure 2F,G). As expected, JWH133 in these cells did not trigger any significant variation of calcium ion levels. The CB_2_R antagonist did not significantly modify the effect of orexin-A (Figure 2F). 

Finally, the mitogen-activated protein kinase (MAPK) pathway was analyzed in single transfected and in cotransfected cells. Data of ERK1/2 phosphorylation obtained by Western blotting and normalized by total ERKs are shown in Figure 2F. JWH133 and orexin-A induced a, respectively, 70% and 50% increase in ERK1/2 phosphorylation over basal phosphorylation levels. The effect was blocked when the two receptors were simultaneously activated. This phenomenon where the agonist of one receptor blocks the activation of the other protomer of the receptor complex is named negative cross-talk, becoming a print to demonstrate the existence of this complex in native tissue. Moreover, pretreatment with the CB_2_R antagonist not only blocked the JWH133-induced effect but also the effect of orexin- A. This cross-antagonism was also found when the OX_1_R antagonist was used. 

### 2.3. CB_2_R and OX_1_R form Tetrameric Complexes via a TM4-TM5 Interface

To characterize the CB_2_R-OX_1_R complex structure, a complementation approach was first used. CB_2_R was fused to the non-fluorescent C-terminal part of the YFP (cYFP hemiprotein) while OX_1_R was fused to the non-fluorescent N-terminal part of the YFP (nYFP hemiprotein). Fluorescence in cells coexpressing these hemiproteins showed bimolecular fluorescence complementation (BiFC), i.e., YFP was reconstituted due to interaction of the receptors. To identify the interacting domains, cells were treated with interfering peptides consisting of the sequence of transmembrane domains (TM) of the receptors fused to a sequence, the TAT cell-penetrating peptide. It has been previously shown that these TAT-derived peptides can be inserted into the plasma membrane and disrupt the interaction of cell surface GPCRs [20]. We found that sequences of the OX_1_R, TAT-TM4 (400 nM), and TAT-TM5 (400 nM) led to a significant decrease in the YFP reconstitution. Similar results were obtained with TAT-TM4 (400 nM) and TAT-TM5 (400 nM) when using CB_2_R TM sequences. These results indicate that the CB_2_R-OX_1_R complex has a TM4-TM5 interface (Figure 3A). To gain insight into the structure of the complex, we explored the possibility that CB_2_R and OX_1_R could form tetrameric complexes. We attempted to obtain BRET in cells expressing a constant amount of the “hemiproteins” OX_1_R-nRLuc and OX_1_R-cRluc and increasing amounts of the “hemiproteins” CB_2_R-nYFP and CB_2_R-cYFP. A hyperbolic BRET signal saturation curve was obtained (BRET_max_ 50 ± 2 mBU; BRET_50_ 50 ± 10 mBU), indicating the formation of tetramers formed by two CB_2_Rs and two OX_1_Rs (Figure 3B).

It was further investigated whether the heteromer print disappeared when disrupting the heteromeric complex using TAT-TM peptides. Assays of cAMP level determination were performed in HEK-293T cells coexpressing the two receptors and treated with TAT-TM peptides. Interestingly, it was observed that the heteromer print disappeared in the presence of TAT-TM4 (400 nM) or TAT-TM5 (400 nM) (CB_2_R sequence) but not when other TAT-TM peptides were used. These two peptides prevented the potentiation of the effect of the CB_2_R agonist-induced by the OX_1_R antagonist (Figure 3E). The action of these peptides was specific, as the TAT-TM2 (400 nM) peptide or vehicle were ineffective. 

Finally, we investigated which G alpha proteins were coupled to the macromolecular CB_2_R-OX_1_R complex. Experiments were performed in HEK-293T cells coexpressing the two receptors and pretreated with either cholera toxin (CTX; 100 ng/mL), which alters G_s_-mediated signaling, pertussis toxin (PTX; 10 ng/mL), which alters G-mediated signaling, or the G_q_ inhibitor, YM254890 (1 µM), which blocks G_q_-mediated signaling and was also used as a further control. In calcium mobilization assays, the effect of orexin-A was partially blocked by PTX and, also, by YM254890 (Figure 3C), suggesting the occurrence of one G_i_ and one G_q_ protein in the heteromeric complex. In cAMP determination assays, PTX treatment completely blocked the decrease in cAMP upon treatment with orexin-A or JWH133 (Figure 3D). Thus, it may be assumed that the tetramer contains at least one G_i_ protein. Furthermore, the fact that CTX did not affect any of the agonistic effects indicates that the complex does not couple to G_s_ proteins.

### 2.4. Blockade of OX_1_R Potentiates CB_2_R Function in the Microglia from APP_Sw/Ind_ Mice

Different studies have shown an important increase in CB_2_R expression in activated microglia compared to resting cells [21]. Accordingly, the expression and function of the CB_2_-OX_1_ receptor complex in the microglia of the Alzheimer’s model and in control mice was determined. To characterize the CB_2_-OX_1_ receptor complex function in microglial cells, cAMP level determination assays were carried out. In resting microglia from control mice, we observed that both JWH133 and orexin-A induced a significant effect that was non-additive upon receptor coactivation. Interestingly, in the presence of the OX_1_R antagonist, JWH133 induced a more marked effect (Figure 4A). When analyzing microglia from APP_Sw/Ind_ mice, both JWH133 and orexin-A provoked a significant decrease over forskolin-induced cAMP levels (Figure 4B) that was non additive in coactivation; such a negative crosstalk was not observed in control microglia. However, pretreatment with the OX_1_R antagonist strongly potentiated CB_2_R signaling, as observed in transfected HEK-293T cells, indicating again that the blockade of OX_1_R potentiates CB_2_R-induced signaling. A cross-antagonism was also found when the CB_2_R antagonist was used; SR144528 blocked the effect triggered by both JWH133 and orexin-A. 

Analysis of the MAPK signaling pathway led to the finding that coactivation with orexin-A and JWH133 induced a lower effect than that observed by activation of one of the receptors, thus, negative crosstalk was stronger in the microglia from control mice but also observed in APP_Sw/Ind_ mice (Figure 4C,D). Interestingly, in APP_Sw/Ind_ mice microglia the CB_2_R antagonist potentiated the OX_1_R-induced function. This phenomenon was not observed in control miceor in transfected HEK-293T cells and could be explained due to a differential expression of CB_2_R in activated microglia.

Finally, by PLA the formation of CB_2_-OX_1_ receptor complexes in control and APP_Sw/Ind_ mice microglia was demonstrated. Interestingly, the expression of the heteromer was two-fold higher in the microglia from the AD model than in the microglia from control animals (Figure 4E). The important increase in the CB_2_-OX_1_ complexes in APP_Sw/Ind_ mice microglia could explain the functional differences observed in cAMP levels and MAPK phosphorylation assays.

## 3. Discussion

The interaction of two receptors that are expressed in microglia opens new perspectives for the regulation of the activation of these cells. In addition, the interaction between CB_2_R and OX_1_R is relevant for understanding the mechanisms underlying orexigenic and cannabinoid functional interactions at the CNS level. A recent study has shown by ligand-binding assays that the phytocannabinoid CBD can bind OX_1_R at low micromolar range where it acts as an antagonist by decreasing calcium mobilization [22]. This result does not become clear evidence of a direct relationship between the cannabinoid and the orexinergic systems because this compound also binds to other receptors such as serotonin receptors [23]. Another study in the amygdala shows that 2-arachidonoylglycerol acting on the CB_2_R reverts the fear extinction deficits induced by orexin-A. This finding seems a consequence of a higher 2-arachidonoylglycerol production by the action of orexin-A. Importantly, the effect of the peptide correlated with an increase in the expression of the CB_2_R in microglial cells [24]. The cannabinoid receptor has been proposed as a target for neuroprotection by regulating microglial activation and polarization. The mechanisms favoring microglia polarization to a neuroprotectivephenotype, also known as M2, are not known, although there are GPCRs that may regulate their production. On the one hand, cannabinoids acting on the CB_2_R may afford neuroprotection and prevent the progression of neurodegenerative diseases (see [25,26] and references therein). On the other hand, the CB_2_R is upregulated in activated microglia. The results afforded by this manuscript demonstrate that the OX_1_R antagonist, SB334867, not only blocks the orexin-A-induced effect, but also potentiates cannabinoid CB_2_R function in cAMP and MAPK signaling pathways. Moreover, this effect is potentiated in activated microglia compared to resting cells. In this sense, OX_1_R antagonists could become a new therapeutic target to consider in AD and other neurodegenerative diseases that trigger with microglia activation.

In the periphery, at the level of the intestinal barrier, orexin-A regulates neuroinflammation by acting on enterocytes and/or microglia [27]. In another of the few papers on orexin-A actions on activated microglia it was reported that the peptide favors the production and release of neuroprotective factors [28]. In animal models, orexin-A also affords protection against neuroinflammation occurring after brain ischemic insult [29]. The hypothesis underlying the research described in this paper was based on the possible co-occurrence and potential interaction of CB_2_R and OX_1_R receptors in microglia. The results confirm that CB_2_R and OX_1_R may interact in both a heterologous expression system and in primary microglia from mouse brain. 

Unfortunately, the 3D structure of heteromers formed by two class A GPCRs has not yet been solved. However, several determined structures display crystallographic interfaces potentially compatible with physiological interactions. Reliable models that have been deciphered using the approach of interfering peptides employed in this study, such as the adenosine A_1_ and A_2A_ heterotetramer coupled to one G_s_ protein and to one G_i_ protein [30,31,32] Previous studies have demonstrated that OX_1_R can form complexes with other proteins with the same TM4-5 interface; a clear example is the heteromeric complex between the apelin receptor APJ and OX_1_R [33]. 

To have a framework to understand the functional regulation within the CB_2_R-OX_1_R hetereomer, we created two structural models of the TM4/5 interface using experimentally determined structures and molecular modeling (see Section 4). The models reveal that the orthosteric binding pockets of CB_2_R and OX_1_R are connected by a network of aromatic residues. It turns out that both the CB_2_R and the OX_1_R antagonists used in the present study place groups near TM5 that trigger a larger opening of the extracellular part of this helix than other antagonists [34,35]. Although the study of this network is beyond the scope of the present work, it is tempting to speculate that an antagonist of one receptor could concertedly push the aromatic residues in this network bringing changes to the orthosteric pocket of the second receptor leading to either cross-activation or cross-inactivation. Figure 5A shows how the SB334867 could affect the signal triggered by the CB_2_R agonist JWH133. Figure 5B shows how the CB_2_R antagonist SR144528 could affect the signal triggered by the endogenous OX_1_R agonist orexin-A. We propose that the opening of TM5 in either case, required to accommodate an antagonist, would modulate the interface. The fact that TM5 exhibits different orientations with different ligands (Figure 5C) could be the basis of the opposite effects of the antagonists (cross-antagonism vs. positive modulation). Thus, the OX_1_R antagonist SB334867 allosterically stabilizes the active form of CB_2_R, potentiating the action induced by the agonist JWH133. By contrast, the CB_2_R antagonist SR144528 allosterically stabilizes an inactive state of the OX_1_R that hinders OXA binding, probably because the second extracellular loop that connects TMs 4 and 5 occludes its access.

Our heterodimeric models in Figure 4 imply the binding of a single G-protein (Appendix A). It is possible that both receptors bind a G-protein (G_i_ or G_q_) in an active complex, but it is not possible that both bind it simultaneously because they would clash sterically. This dimeric model already provides the framework to understand the allosteric regulations by different protomers. Further work would be required to understand the benefits of having a heterotetramer compared to a heterodimer. The results in transfected HEK-293T cells have demonstrated the formation of tetrameric complexes with one homodimer of CB_2_R and one homodimer of OX_1_R able to signal through both G_i_ and a G_q_. A tetramer could eventually allow the binding of G_q_ to one OX_1_ protomer and G_i_ to another. In fact, a tetramer assuming symmetric TM1 interfaces could simultaneously bind up to three G-proteins (Appendix A).

Some studies have demonstrated the exchange of G protein coupling to GPCRs when forming complexes in specific tissues. One of the first examples is that described by M Glass and collaborators, where cotreatment with a CB_1_R agonist and a dopaminergic D_2_R agonist increased cAMP levels in striatal neurons [36]. Thus, this indicates that two receptors that typically couple to the G_i_ protein can couple to a G_s_ protein when forming CB_1_R-D_2_R complexes in striatal neurons. Another example is D_1_R and D_2_R mediated signaling via the established G_s/olf_ and G_i/o_, respectively. This is abolished when the heterodimer is stimulated, leading to new signal transduction by G_q_/11 in the striatum [37]. In this sense, we have demonstrated that the newly described complex functions through the OX_1_R and CB_2_R characteristic proteins, a G_q_ and a G_i_, respectively, and, although the heteromer permits the OX_1_R to signal via G_i_ as well. 

## 4. Materials and Methods

### 4.1. Reagents

JWH133, SR144528, orexin-A, SB334867 and YM 254890 were purchased from Tocris Bioscience (Bristol, UK). Forskolin, cholera toxin from *Vibrio cholerae* (CTX) and pertussis toxin (PTX) from *Bordetella pertussis* were purchased from Sigma-Aldrich (St. Louis, MO, USA), and the G_q_ inhibitor, YM254890, was obtained from Focus Biomolecules (Plymouth Meeting, PA, USA).

### 4.2. Cell Isolation and Culturing

HEK-293T cells (batch 70022180) were acquired from the American Type Culture Collection (ATCC). Cells were amplified and frozen in liquid nitrogen in several aliquots. Cells from each aliquot were used until passage 18. HEK-293T cells were grown in Dulbecco’s Modified Eagle’s Medium (DMEM) (Gibco) supplemented with 2 mM L-glutamine, 100 μg/mL sodium pyruvate, 100 U/mL penicillin/streptomycin, MEM non-essential amino acids solution (1/100) and 5% (*v*/*v*) heat-inactivated fetal bovine serum (FBS) (all supplements were from Invitrogen, Paisley, Scotland, UK) and maintained at 37 °C in a humid atmosphere of 5% CO_2_.

CD-1 strain mice handling, sacrifice, and further experiments were conducted according to the guidelines set in Directive 2010/63/EU of the European Parliament and the Council of the European Union that is enforced in Spain by National and Regional organizations; the 3R rule (replace, refine, reduce) for animal experimentation was also considered. Primary cultures of microglia were obtained from 2–3-day-old pups. Cells were isolated as described in [41] and plated at a confluence of 40,000 cells/0.32 cm^2^. Briefly, the samples were dissected, carefully stripped of their meninges and digested with 0.25% trypsin for 20 min at 37 °C. Trypsinization was stopped by repeated washes with Hanks′ balanced salt solution (HBSS composition: 1.26 mM CaCl_2_, 5 mM KCl, 0.44 mM KH_2_PO_4_, 0.5 mM MgCl_2_, 0.4 mM MgSO_4_, 137 mM NaCl, 0.34 mM Na_2_HPO_4_ and 10 mM Hepes, pH: 7.4). Cells were brought to a single cell suspension by repeated pipetting followed by passage through a 100 µm-pore mesh. Cells were then resuspended in supplemented DMEM and seeded at a density of 3.5 × 10^5^ cells/mL in 6-well plates or 96-well plates for functional assays and in twelve-well plates for immunocytochemistry or PLA assays. Cultures were maintained at 37 °C in a humidified 5% CO_2_ atmosphere. The medium was replaced every 4–5 days. Immunodetection of specific markers (CD-11b) showed that microglia preparations contained at least 98% microglial cells [42]

### 4.3. Cell Transfection

HEK-293T cells were transiently transfected with the corresponding cDNA by the PEI (PolyEthylenImine, Sigma-Aldrich) method. Briefly, the corresponding cDNA diluted in 150 mM NaCl was mixed with PEI (5.5 mM in nitrogen residues), also prepared in 150 mM NaCl for 10 min. The cDNA-PEI complexes were transferred to HEK-293T cells and were incubated for 4 h in a serum-starved medium. The medium was then replaced by a fresh supplemented culture medium and cells were maintained at 37 °C in a humid atmosphere of 5% CO_2_. Forty-eight hours after transfection, cells were washed, detached, and resuspended in the assay buffer.

### 4.4. Fusion Proteins and Expression Vectors

Plasmids encoding for CB_2_R-YFP, CB_2_R-nYFP, CB_2_R-cYFP, OX_1_R-Rluc, OX_1_R-nRluc, OX_1_R-cRluc, and OX_1_R-nYFP proteins were available in our laboratory. 

Amplified cDNA fragments of the receptor were subcloned to be in-frame with restriction sites of pRluc-N1, pEYFP-N1, nYFP-pcDNA3.1, cYFP-pcDNA3.1, nRluc-pcDNA3.1 and cRluc-pcDNA3.1 vectors to provide plasmids that express proteins fused to the C-terminal end of Rluc, YFP, nRluc or cRluc.

### 4.5. Transgenic Alzheimer’s Disease (AD) Animal Model

APP_Sw,Ind_ transgenic mice (line J9; C57BL/6 background) expressing human APP695 harboring the FAD-linked Swedish (K670N/M671L) and Indiana (V717F) mutations under the PDGF promoter were obtained by crossing APP_Sw,Ind_ to non-transgenic (WT) mice [43]. APP_Sw,Ind_-derived embryos or pups, individually genotyped and divided into “APP_Sw,Ind_” and “control”, were used for preparing primary cultures. Animal care and experimental procedures were in accordance with European and Spanish regulations (86/609/CEE; RD1201/2005). Mice were handled, as per law, by personnel with the ad hoc permit (issued by the Generalitat de Catalunya), which allows animal handling for research purposes. 

### 4.6. Immunocytochemistry 

HEK-293T cells were seeded on glass coverslips in 12-well plates. Twenty-four hours later, cells were transfected with CB_2_R-YFP cDNA (1 μg) and/or OX_1_R-Rluc cDNA (1 μg). Forty-eight hours after, cells were fixed in 4% paraformaldehyde for 15 min and washed twice with PBS containing 20 mM glycine before permeabilization with PBS-glycine containing 0.2% Triton X-100 (5 min incubation). Cells were blocked during 1 h with PBS containing 1% bovine serum albumin. HEK-293T cells were labeled with a mouse anti-Rluc antibody (1/100, MAB4400, Millipore, Merck, Darmstadt, Germany) and subsequently treated with a Cy3-conjugated anti-mouse (1/200, 715-166-150 (red), Jackson ImmunoResearch, St. Thomas Place, UK) IgG secondary antibody (1 h each). The CB_2_R-YFP expression was detected by the YFP’s own fluorescence. Nuclei were stained with Hoechst (1/100 from stock 1 mg/mL; Sigma-Aldrich). Samples were washed several times and mounted with 30% Mowiol (Calbiochem). Images were obtained in a Zeiss LSM 880 confocal microscope (ZEISS, Germany) with the 63X oil objective.

### 4.7. Bioluminescence Resonance Energy Transfer (BRET) Assay and BRET with BiFC Assays

For the BRET assay, HEK-293T cells were transiently cotransfected with a constant amount of cDNA encoding for OX_1_R-Rluc (1 μg) and with increasing amounts of cDNA corresponding to CB_2_R-YFP (0.4 to 1.6 μg). As negative control, HEK-293T cells were transiently cotransfected with a constant amount of cDNA encoding for GABA_B_-Rluc (0.4 μg) and with increasing amounts of cDNA corresponding to CB_2_R-YFP (0.8 to 5 μg). For the BRET assays with Bimolecular fluorescence complementation (BiFC), HEK-293T cells were transiently cotransfected with a constant amount of cDNA encoding for proteins fused to Rluc hemiproteins (nRluc, cRluc), OX_1_R-nRluc8 (1.5 μg) and OX_1_R-cRluc8 (1.5 μg) and with increasing amounts of the cDNA corresponding to proteins fused to YFP hemiproteins (nYFP, cYFP), CB_2_R-nYFP Venus (0.3–2.5 µg), or CB_2_R-cYFP Venus (0.3–2.5 µg). To control the cell number, the sample protein concentration was determined using a Bradford assay kit (Bio-Rad, Munich, Germany) using bovine serum albumin (BSA) dilutions as standards. To quantify fluorescent proteins, cells (20 μg of total protein) were distributed in 96-well microplates (black plates with a transparent bottom) and fluorescence was read in a Fluostar Optima Fluorimeter (BMG Labtech, Ofenburg, Germany) equipped with a high-energy xenon flash lamp using a 10-nm bandwidth excitation filter at 485 nm. For BRET measurements, the equivalent of 20 μg of total protein cell suspension was distributed in 96-well white microplates with a white bottom (Corning 3600, Corning, NY). For BRET measurements, the equivalent to 20 µg cell suspension was distributed in 96-well microplates (white plates, Porvair, Leatherhead, UK) and 5 µM coelenterazine H was added (PJK GMBH, Kleinblittersdorf, Germany). One minute after coelenterazine H addition, the readings were collected using a Mithras LB 940 (Berthold, Bad Wildbad, Germany), which allowed the integration of the signals detected in the short-wavelength filter at 485 nm (440–500 nm) and the long-wavelength filter at 530 nm (510–590 nm). To quantify receptor-Rluc expression, luminescence readings were collected 10 min after the addition of 5 µM coelenterazine H. The net BRET is defined as [(long-wavelength emission)/(short-wavelength emission)]-C_f_, where C_f_ corresponds to [(long-wavelength emission)/(short-wavelength emission)] for the Rluc construct expressed alone in the same experiment. Data in BRET curves that depict an equilateral hyperbola were fitted by a non-linear regression equation using GraphPad Prism software (San Diego, CA, USA). BRET values for specific interactions are given as milli BRET units (mBU: 1000 × net BRET). For BiFC assays, HEK-293T cells were transiently transfected with a constant amount of cDNA encoding for proteins fused to nVenus (OX_1_R-nYFP) or cVenus (CB_2_-cYFP) and incubated for 4 h in complete DMEM containing the interfering TAT peptides (with similar sequences to those in TM1 to TM7 for CB_2_R or TM4, TM5 or TM7 for OX_1_R). YFP resulting from complementation was detected by placing cells (20 μg protein) in 96-well microplates (black plates with a transparent bottom) and reading the fluorescence in a Fluostar Optima Fluorimeter (BMG Labtech, Offenburg, Germany) using a 30-nm bandwidth excitation filter (485 nm).

### 4.8. TAT-TM Peptides 

Peptides with the sequence of the TM of CB_2_R and OX_1_R fused to the HIV TAT peptide (YGRKKRRQRRR) were used as oligomer-disrupting molecules (synthesized by Genemad Synthesis Inc. San Antonio, TX, USA). The cell penetrating TAT peptide allows for the intracellular delivery of fused peptides [20]. The TAT-fused TM peptide can then be inserted effectively into the plasma membrane because of the penetration capacity of the TAT peptide and the hydrophobic property of the TM moiety [44]. To obtain the right orientation of the inserted peptide, the HIV-TAT peptide was fused to the C-terminus or to the N-terminus as indicated: 

TAVAVLCTLLGLLSALENVAVLYLIL-YGRKKRRQRRR for CB_2_R TM1,

YGRKKRRQRRR-YLFIGSLAGADFLASVVFACSFVNF for CB_2_R TM2,

AVFLLKIGSVTMTFTASVGSLLLTAI-YGRKKRRQRRR for CB_2_R TM3,

YGRKKRRQRRR-ALVTLGIMWVLSALVSYLPLMGW for CB_2_R TM4,

YLLSWLLFIAFLFSGIIYTYGHVLW-YGRKKRRQRRR for CB_2_R TM5,

YGRKKRRQRRR-TLGLVLAVLLICWFPVLALMAH for CB_2_R TM6,

AFAFCSMLCLINSMVNPVIYAL-YGRKKRRQRRR for CB_2_R TM7,

YGRKKRRQRRR-ILGIWAVSLAIMVPQAAVME for OX_1_R TM4, 

SSFFIVTYLAPLGLMAMAYFQIF-YGRKKRRQRRR for OX_1_R TM5, 

YASFTFSHWLVYANSAANPIIYNF-YGRKKRRQRRR for OX_1_R TM7.

### 4.9. cAMP Level Determination 

The analysis of cAMP levels was performed in HEK-293T cells cotransfected with the cDNA for CB_2_R (1.5 μg) and the cDNA for the OX_1_R (1.5 μg), in primary microglia and primary neurons prepared from wild-type mice or the transgenic APP_Sw/Ind_ AD mice model. In the case of HEK-293T cells, when indicated, were treated overnight with the required TAT peptides TM2, TM4, and TM5 (0.4 µM) or with 10 ng/mL pertussis toxin (PTX; ref: P7208-50UG, Sigma-Aldrich, over-night), 100 ng/mL cholera toxin (CTX; ref:C8052-5M, Sigma-Aldrich, 2 h) or 1µM G_q_ inhibidor (YM 254890, 2 h). Two hours before the experiment, the medium was replaced by serum-starved DMEM medium. Cells growing in a medium containing 50 μM zardaverine were distributed in 384-well microplates (2000 HEK-293T or 4000 primary cells per well) treated at 37 °C with selective antagonists (1 µM SB334867 or 1 µM SR144528) for 15 min prior stimulation with OX_1_R and/or CB_2_R agonists (100 nM orexin-A and/or 100 nM JWH133); 15 min after agonists treatment, cells were treated (15 min) with 0.5 μM forskolin or vehicle. Homogeneous time-resolved fluorescence energy transfer (HTRF) measurements were performed using the Lance Ultra cAMP kit (PerkinElmer). Fluorescence at 665 nm was analyzed on a PHERAstar Flagship microplate reader equipped with an HTRF optical module (BMG Labtech). A standard curve for cAMP was obtained in each experiment.

### 4.10. Detection of Cytoplasmic Calcium Levels

HEK-293T cells were cotransfected with the cDNA for the protomers of the CB_2_R (1.5 μg), the cDNA for OX_1_R (1.5 μg) and with the cDNA for the GCaMP6 calcium sensor (1 μg) by the PEI method (Section 4.3). Forty-eight hours after transfection, HEK-293T cells plated in 6-well black, clear bottom plates, were incubated with Mg^2+^-free Locke’s buffer (154 mM NaCl, 5.6 mM KCl, 3.6 mM NaHCO_3_, 2.3 mM CaCl_2_, 5.6 mM glucose, 5 mM HEPES, 10 μM glycine, pH 7.4). Online recordings were performed right after the addition of agonists. When indicated, cells were pre-treated with receptor antagonists for 10 min. Fluorescence emission intensity due to complex GCaMP6 was recorded at 515 nm upon excitation at 488 nm on the EnSpire^®^ Multimode Plate Reader for 150 s every 5 s at 100 flashes per well.

### 4.11. Extracellular Signal-Regulated Kinase (ERK) Phosphorylation Assays

HEK-293T cells were transfected with the cDNA encoding for CB_2_R and for OX_1_R. Two to four hours before initiating the experiment, the culture medium was replaced by serum-starved DMEM medium. Cells were treated at 37 °C with selective antagonists (1 µM SB334867 or 1 µM SR144528) for 10 min prior to stimulation with OX_1_R and/or CB_2_R agonists (100 nM orexin-A and/or 100 nM JWH133) for 7 min. Cells were then placed in an ice-water bath and washed twice with cold PBS and lysed with the addition of ice-cold lysis buffer (50 mM Tris-HCl pH 7.4, 50 mM NaF, 150 mM NaCl, 45 mM ß-glycerolphosphate, 1% Triton X-100, 20 µM phenyl-arsine oxide, 0.4mMNaVO4 and the protease inhibitor mixture (MERK, St. Louis, MO, USA)). Cellular debris were removed by centrifugation at 12,000 rpm for 10 min at 4 °C, and protein was adjusted to 1 mg/mL by the bicinchoninic acid method (ThermoFisher Scientific, Waltham, MA, USA) using a commercial bovine serum albumin dilution (BSA) (ThermoFisher Scientific) for standardization. Finally, cells were denatured by placing them at 100 °C for 5 min. ERK1/2 phosphorylation were determined by western blot. Equivalent amounts of protein (20 µg) were subjected to electrophoresis (10% SDS-polyacrylamide gel) and transferred onto PVDF membranes (Immobilon-FLPVDF membrane, MERK, St. Louis, MO, USA) for 90 min. The membranes were then blocked for 1 h at room temperature (constant shaking) with Odyssey Blocking Buffer (LI-COR Biosciences, Lincoln, NE, USA) and labelled with a mixture of primary mouse anti-phospho-ERK1/2 antibody (1:2000, MERK, Ref. M8159), primary rabbit anti-ERK1/2 antibody (1:40,000, MERK, Ref. M5670), which recognizes both phosphorylated and unphosphorylated ERK1/2 overnight at 4 °C with shaking. The membranes were then washed three times with PBS containing 0.05% tween and visualized by the addition of a mixture of IRDye 800 anti-mouse antibody (1:10,000, MERK, Ref. 926-32210) and IRDye 680 anti-rabbit antibody (1:10,000, MERK, Ref. 926-68071) for 2 h at room temperature. Membranes were washed three times with PBS-tween 0.05% and once with PBS and left to dry. Bands were analyzed using an Odyssey infrared scanner (LI-COR Biosciences).

Band densities were quantified using Fiji software, and the level of phosphorylated ERK1/2 was normalized using the total ERK1/2 protein band intensities. Results obtained are represented as the percent over basal (non-stimulated cells).

To determine the ERK1/2 phosphorylation in primary cultures, 50,000 cells/well were plated in transparent 96-well microplates and kept in the incubator for 14 days. Two hours before initiating the experiment, the medium was substituted by serum-starved DMEM medium. Cells were then treated at 37 °C with selective antagonists (1 µM SB334867 or 1 µM SR144528) for 7 min prior stimulation (25 °C) with OX_1_R and/or CB_2_R agonists (100 nM orexin-A and/or 100 nM JWH133). ERK1/2 phosphorylation was determined using an AlphaScreen^®^ SureFire^®^ kit (Perkin Elmer) following the instructions of the supplier and using an EnSpire^®^ Multimode Plate Reader (PerkinElmer, Waltham, MA, USA).

### 4.12. Proximity Ligation Assay (PLA) 

Direct interaction between CB_2_R and OX_1_R was detected using the Duolink in situ PLA detection Kit (OLink; Bioscience, Uppsala, Sweden) following the instructions of the supplier. Primary neurons and microglia were grown on glass coverslips, fixed in 4% paraformaldehyde for 15 min, washed with PBS containing 20 mM glycine to quench the aldehyde groups and permeabilized with the same buffer containing 0.05% Triton X-100 (20 min). Samples were then successively washed with PBS. After 1 h of incubation at 37 °C with the blocking solution in a pre-heated humidity chamber, cells were incubated overnight in the antibody diluent medium with a mixture of equal amounts of rabbit anti-CB_2_R (ab230791, Abcam, Cambridge,UK) directly coupled to a plus DNA strand (obtained following the instructions of the Sigma-Aldrich supplier, ref: DUO92010-1KT) (1/100) and rabbit anti-OX_1_R (ab83960, Abcam Cambridge,UK) directly coupled to a minus DNA strand (obtained following the instructions of the Sigma-Aldrich supplier, ref: DUO92009-1KT) (1/100). Ligation and amplification were conducted as indicated by the supplier. Samples were mounted using the mounting medium with Hoechst (1/100; Sigma-Aldrich, St. Louis, MO, USA) to stain nuclei. Samples were observed in a Zeiss 880 confocal microscope (Carl Zeiss, Oberkochen, Germany) equipped with an apochromatic 63× oil immersion objective (N.A. 1.4) and 405 nm and 561 nm laser lines. For each field of view, a stack of two channels (one per staining) and four Z stacks with a step size of 1 µm were acquired. The number of neurons or microglia containing one or more red spots versus total cells (blue nucleus) was determined, and Bonferroni’s multiple comparison post hoc test was used to compare the values (red dots/cell).

### 4.13. Statistical Analysis 

GraphPad Prism 8 software (San Diego, CA, USA) was used for data fitting and statistical analysis. One-way ANOVA followed by post hoc Bonferroni’s test was used when comparing multiple values. From PLA confocal images the number of red dots/cell was quantified using the Andy’s algorithm Fiji’s plug-in [45].

### 4.14. Molecular Modelling 

We used experimentally determined structures to model the monomeric structures of both CB_2_R and OX_1_R in inactive and active states. The complex CB_2_R/JWH133 was modeled based on the active structure of CB_2_R bound to the structurally related hexaydrocannabinol (PDB id 6KPF; [46]). The complex CB_2_R/SR144528 was modeled based on the inactive structure of CB_2_R bound to antagonist AM10257 (PDB id 5ZTY; [47], which also shares the central pyrazole ring located in the same position [34].The complex OX_1_R/SB334867 has been determined experimentally (PDB id 6TQ7; [35]. The complex (N-terminal part of) OX_1_R/orexin-A was modeled based on the OX_2_R/orexin-B complex (PDB id 7L1U; [48].

We used DIMERBOW [49]) to explore for possible TM4/5 interfaces to model the CB_2_R/OX1 dimer. We selected the structure of the 5-HT_2C_ receptor (PDB id 6BQG; [50]), which reported a compact symmetric TM4/5 interface. In turn, we superposed one CB_2_R and one OX_1_R monomeric complex to the 5-HT_2C_ dimer to obtain the dimeric complexes CB_2_R/JWH133-OX_1_R/SB334867 and CB_2_R/SR144528-OX_1_R/orexin-A. Because these structures showed clashes between helices TMs 4–5 of both protomers, we created new monomeric models incorporating the TMs 4 and 5 and proximal parts of the second extracellular loop in the conformation observed in the 5-HT_2C_ receptor. All homology models of monomers were created using Modeller 10.3 [51]. All complexes (monomeric and dimeric) were energy minimized with AMBER22 using the ff19SB forcefield [52]. Ligands were parametrized using GAFF2 [53].

## Figures and Tables

**Figure 1 ijms-23-12801-f001:**
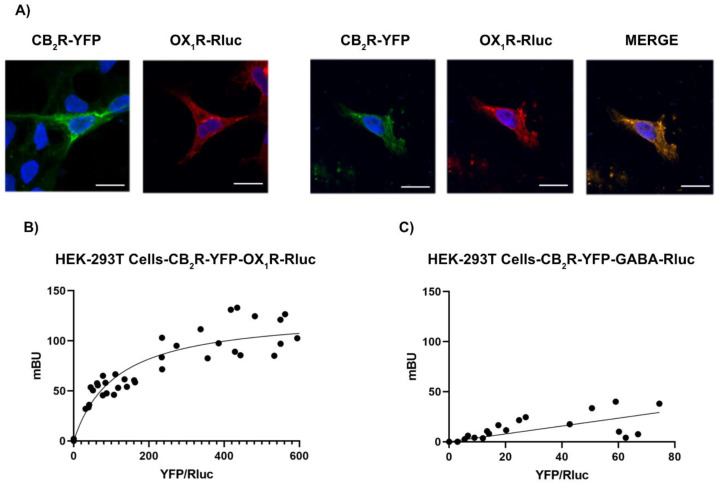
The OX_1_R and CB_2_R interact in a heterologous expression system. (**A**) Immunocytochemistry assays were performed in HEK-293T cells expressing CB_2_R-YFP (1 μg cDNA), which was detected by its own fluorescence (green) and/or OX_1_R-Rluc (1 μg cDNA) that was detected by a mouse monoclonal anti-Rluc antibody and a secondary Cy3-conjugated anti-mouse IgG antibody (red). Colocalization is shown in yellow. Cell nuclei were stained with Hoechst (blue). Scale bar: 15 μm. (**B**) BRET assays were performed in HEK-293T cells transfected with a constant amount of cDNA for OX_1_R-Rluc (0.4 μg) and increasing amounts of cDNA for CB_2_R-YFP (0.4 to 1.6 μg) or, as negative control (**C**), a constant amount of GABA_B_-Rluc (0.4 μg cDNA) and increasing amounts of CB_2_R- YFP (0.4 to 1.6 μg cDNA). Values correspond to six independent experiments.

**Figure 2 ijms-23-12801-f002:**
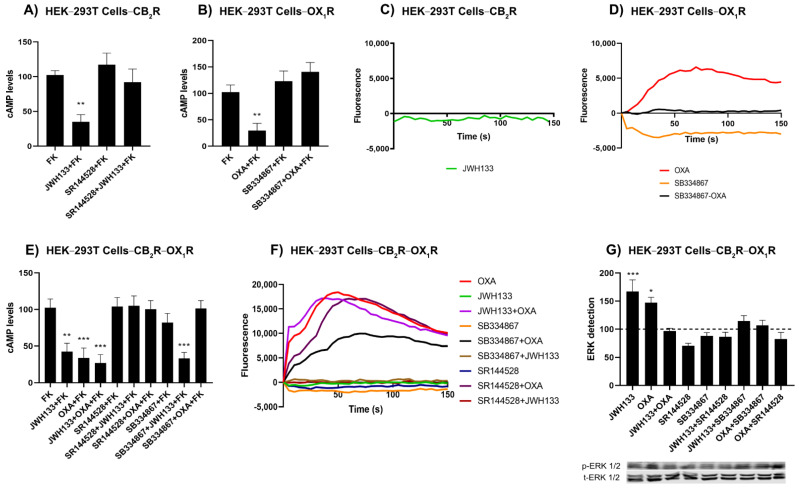
Functionality of OX_1_R-CB_2_R heteromer in HEK-293T cells. (**A**,**B**,**E**) HEK-293T cells were transfected with the cDNA for the CB_2_ receptor (1 μg) (**A**), for the OX_1_ receptor (1 μg) (**B**) or both (**E**) and were pretreated with selective receptor antagonists (1 μM SB334867 for OX_1_R or 1 μM SR144528 for CB_2_R) or vehicle and treated with selective agonists (100 μM Orexin-A for OX_1_R and/or 100 nM JWH133 for CB_2_R) followed by 0.5 μM forskolin stimulation (15 min). Values are the mean ± S.E.M. of six independent experiments performed in triplicates. One-way ANOVA followed by Bonferroni’s multiple comparison post hoc test were used for statistical analysis (*** p* < 0.01, *** *p* < 0.001) versus FK condition. (**C**,**D**,**F**) HEK-293T cells were transfected with the cDNA for the GCaMP6 calcium sensor (1 μg) and for the CB_2_ receptor (1 μg) (**C**), for the OX_1_ receptor (1 μg) (**D**) or both receptors (**F**). Receptors were activated using selective agonists (100 nM orexin-A for OX_1_R and/or 100 nM JWH133 for CB_2_R). When indicated, cells were pretreated with selective antagonists (1 μM SB334867 for OX_1_R or 1 μM SR144528 for CB_2_R). Cytosolic calcium readings were collected and data are the mean ± S.E.M. from six independent experiments. (**G**) ERK1/2 phosphorylation was determined in HEK-293T cells expressing the CB_2_ receptor (1 μg cDNA) and the OX_1_ receptor (1 μg cDNA). Results are expressed in percentage with respect to basal condition. Values are the mean ± S.E.M. of five independent experiments performed in triplicates. A resentative Western blot is shown (bottom). p-ERK1/2: phosphorylated ERKs; t-ERK1/2: total ERKs. ** p* < 0.05, *** *p* < 0.001. One-way ANOVA followed by Bonferroni’s multiple comparison post hoc test were used for statistical analysis * *p* < 0.05 versus basal condition in ERK1/2 phospohorylation assays or versus FK condition in cAMP determination assays (dashed line).

**Figure 3 ijms-23-12801-f003:**
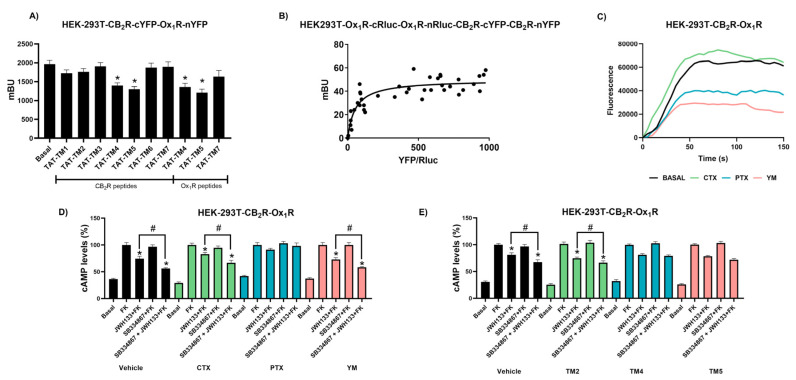
Transmembrane domains involved in the OX_1_R-CB_2_R interaction. (**A**) Bimolecular complementation experiments were determined in HEK-293T cells coexpressing OX_1_R-nYFP (1.5 µg) and CB_2_R-cYFP (1.5 µg) and treated for 4 h with interfering peptides having sequences with homologies to transmembrane (TM) domains (0.4 μM). Values represent the mean ± SEM of eight independent experiments. A statistical analysis was performed using a one-way ANOVA followed by Bonferroni’s post hoc test (* *p* < 0.05 versus basal condition). (**B**) BRET assays were determined in cells expressing a constant amount of cDNA for OX_1_R-cRluc (1.5 µg) and OX_1_R-nRluc (1.5 µg) and increasing amounts of cDNA for CB_2_R-cYFP (0.3–2.5 µg) and CB_2_R-nYFP (0.3–2.5 µg). (**C**) HEK-293T cells expressing CB_2_R, OX_1_R and the engineered calcium sensor, GCaMP6 (1 μg) were incubated overnight with vehicle or pertussis toxin (PTX; 10 ng/mL), or for 2 h with cholera toxin (CTX; 100 ng/mL) or YM254890 (YM; 100 ng/mL) and exposed to orexin-A (100 nM). Cytosolic calcium readings were collected, and data from a representative experiment are shown. (**D**) HEK-293T cells expressing the receptors were incubated overnight with vehicle or pertussis toxin (PTX; 10 ng/mL), or for 2 h with cholera toxin (CTX; 100 ng/mL) or YM254890 (YM; 100 ng/mL) and exposed to JWH133 (100 nM), SB334867 (1 μM) or both in the presence of forskolin (0.5 μM). (**E**) Cells coexpressing cDNAs for OX_1_R (1.5 µg) and CB_2_R (1.5 µg) were preincubated for 4 h with interfering peptides (0.4 µM), pretreated with CB_2_R antagonist (SB334867 1 µM) for 15 min and then treated for 15 min with the selective CB_2_R agonist, JWH133 (100 nM), in the presence of forskolin before determining cAMP levels. A statistical analysis was performed using a one-way ANOVA followed by Bonferroni’s post hoc test. * *p* < 0.05 versus forskolin treatment # *p* < 0.05 versus JWH133 + FK condition. Values are the mean ± S.E.M. of five different experiments performed in triplicates. Values are expressed as percentage of cAMP accumulation provoked by forskolin (FK) (n =  3, in triplicates). One-way ANOVA followed by Bonferroni’s multiple comparison tests was used for statistical analysis * *p* < 0.05 # *p* < 0.05 versus JWH133 + FK condition.

**Figure 4 ijms-23-12801-f004:**
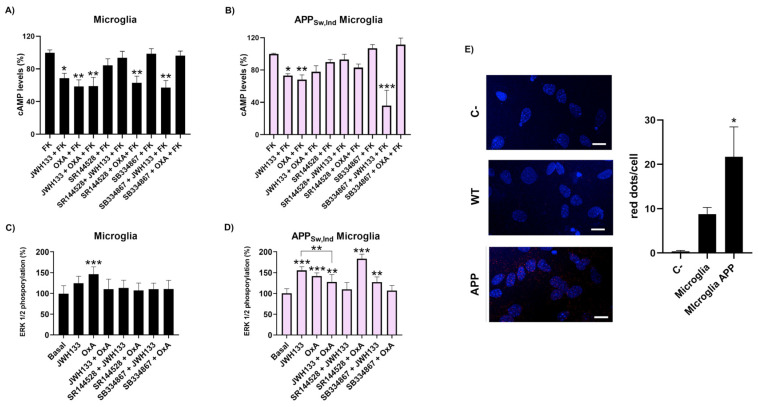
Functionality of CB_2_R and OX_1_R heteromer in the microglia from the APP_Sw,Ind_ mouse. Primary microglia from APP_Sw,ind_ (**A**,**B**) or age-matched control animals (**B**,**D**) were pre-treated with selective receptor antagonists (1 µM SR144528 for CB_2_ or 1 µM SB334867 for OX_1_ receptors) for 15 min and subsequently treated with selective agonists (100 nM JWH133 for CB_2_ or 100 nM orexin-A for OX_1_ receptors) in single or combined treatments. cAMP levels (**A**,**B**) were detected 15 min after forskolin addition. Values are the mean ± S.E.M. of seven different experiments performed in triplicates. One-way ANOVA followed by Bonferroni’s multiple comparison post hoc test was used for statistical analysis (* *p* < 0.05 versus forskolin treatment). (**C**,**D**) ERK1/2 phosphorylation was analyzed using an AlphaScreen^®^ SureFire^®^ kit (Perkin Elmer). Values are the mean ± S.E.M. of five different experiments performed in triplicates. One-way ANOVA followed by Bonferroni’s multiple comparison post hoc test were used for statistical analysis (*** p* < 0.01, *** *p* < 0.001ersus untreated cells). (**E**) A proximity ligation assay (PLA) was performed in primary microglia from APP_Sw,Ind_ transgenic mice or age-matched controls, using specific primary antibodies against CB_2_R or against OX_1_R (1/100). Representative images corresponding to stacks of four sequential planes are shown. Cell nuclei were stained with Hoechst (blue) and receptor complexes appear as red dots. The number of red dots/cell was quantified using Andy’s algorithm Fiji’s plug-in (see Section 4). Scale bar: 15 µm. Values are the mean ± S.E.M. of five different experiments performed in duplicates. One-way ANOVA and Bonferroni’s multiple comparison post hoc test were used for statistical analysis (* *p* < 0.05 versus control).

**Figure 5 ijms-23-12801-f005:**
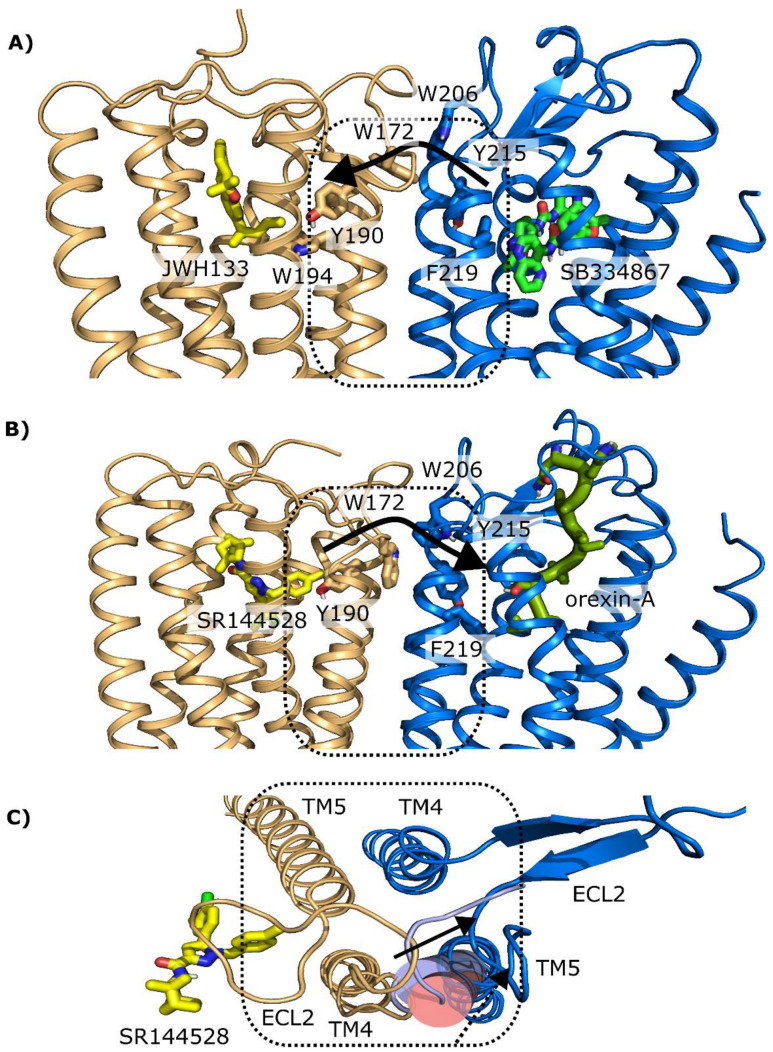
Computational model of the heteromeric TM4/5 CB_2_R-OX_1_R interface. (**A**,**B**) The possible modulation (indicated by the black arrows) of an antagonist to a neighbor receptor through a network of aromatic residues connecting both orthosteric sites, viewed from the membrane. (**A**) the CB_2_R/JWH133-OX_1_R/SB334867 complex; The experimental structure of the OX_1_R antagonist SB334867 revealed the presence of two copies of the ligand in the binding pocket interacting antiparallelly with each other, which resulted in a large ligand volume near TM5. (**B**) The CB_2_R/SR144528-OX_1_R/orexin-A. Similarly, the CB_2_R antagonist SR144528 puts an aromatic ring at the region where most CB ligands place the alkyl chain (such as in JHW133), also towards TM5. The CB_2_R receptor is shown in gold and the OX_1_R receptor in blue. Small-molecule ligands and aromatic side chains connecting the pockets are shown as sticks; orexin-A with a thick green cartoon. Dotted boxes indicate TMs 4 and 5 of both receptors. In (**C**) detail of the CB_2_R/SR144528-OX_1_R/orexin-A complex viewed from the extracellular side superposing different orientations of the extracellular portion of TM5 and proximal region of the second extracellular loop (ECL2) (i) from the OX_2_R/orexin-B complex (PDB id 7L1U, cylinder and loop in light blue), which would clash sterically with the TM4 of the CB_2_R; and (ii) from the OX_1_R/SB334867 complex (PDB id 6TQ7, pink cylinder), which orients in a direction that does not directly face TM4 and better avoids the possible clashes with it TM4. Arrows indicate the movements of TM5 helices towards the TM5 modeled in the heteromer (see Section 4, dark blue) [33]. This complex shows a characteristic signaling where the OX_1_R selective antagonist potentiates the cannabinoid CB_2_R function in cAMP signaling while it blocks the indirect MAPK pathway. These interesting results are also observed in resting microglia. However, in activated microglia, the OX_1_R antagonist blockade became stronger, inducing a higher potentiation of CB_2_R cannabinoid signaling not only in cAMP but also in MAPK signaling. These results can be explained because in primary microglia from the transgenic AD model, the CB_2_-OX_1_ receptor heteromer expression is significantly higher than in equivalent cells from control animals, as shown by PLA. This finding suggests that the heteromer could be a target to combat neurodegeneration. We have previously shown that primary microglia from this AD model have an activated phenotype that is, likely, neuroprotective [38,39,40]. This hypothesis would explain why cognitive deficits do not appear at birth but later in life. Future research should address whether the CB_2_-OX_1_ receptor heteromer is a suitable target to skew microglia to the neuroprotective phenotype.

## Data Availability

Data that may be eventually missing can be obtained from the corresponding author upon reasonable request.

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
