# Peer review of "Antagonization of OX_1_ Receptor Potentiates CB_2_ Receptor Function in Microglia from APP_Sw/Ind_ Mice Model"

_ijms, 2022, doi:10.3390/ijms232112801_

Round 1

Reviewer 1 Report

The manuscript by Raich et al. Deals with the heteromerization fo Ox1 and CB2 receptors focussing in particular on cultured microglia cells from APPSw/Ind mice.

The   working hypothesis deserves attention since it could provide new insights of the pathological cellular events underlying the development of the neurodegenerative disorder, also suggesting new therapeutic targets for its management. Nonetheless, despite the relevance of the results, a deep revision of the manuscript is required to improve its quality and to remove the several speculations or misinterpretations of the data that reduce the relevance of the study.

The introduction is exhaustive and gives a wide overview of the available knowhow concerning the two receptors (OX1 and CB2 receptors) in the CNS also providing few information concerning their role in the Alzheimer’s disease (AD).

The methods are sufficiently described. As to the use of animals at the topo of paragraph 2.5 the authors state that “By the current legislation, obtaining protocol approval is not needed if  animals are sacrificed to obtain a specific tissue”. To the best of my knowledge however the protocol approval is required for transgenic animals as indeed reported   in paragraph 2.6.  Please clarify this point in the section 2.6 or, alternatively, remove the above reported sentence in section 2.5.

Results section

Lines 430-432: not clear why it is expected that JWH133 should not trigger any variation of calcium ion levels, taking into consideration that the CB2 receptor is coupled to the adenylil cyclase which produces cAMP taht in turn reverberate on VOCCs controlling teh influx of calcium ions within cells. Please explain or modify the text.

Line 408: the sentence  “Ox1R can couple to different G proteins; hence we first performed assays assuming that it could couple to Gi protein.”   The statement is redundant since the authors did not discuss any other possibility despite the puzzling results observed in microglia cells from transgenic mice.

Lines 473-475: The HIV-1 protein Tat efficiently permeates membranes and can enter cells and when linked to chemical it favors their internalization within cells. However the protein does not favour the viral internalization, rather it controls the efficiency of the viral transcription. Please modify the statement or remove it.

Lines 533 to 535: the term small is inappropriate unnecessary and must be removed. The authors did not carried out comparative studies with the other experimental control conditions.

Line 539: again, the term counteracted is inappropriate and speculative and suggest a conclusion that is not supported by the results. Looking at the figure it emerge that when concomitantly added the two agonists cause a reduction that does not differ from that elicited by the agonist alone as well as from the FK alone. The sentence “such a negative cross-talk  that was not observed in control microglia.” is not supported by the data and sounds speculative.

Lines 533- 544 The results that would support the negative (?) cross-talk linking the two receptors are confusing and over-speculative. The sentence must be modified to describe the results as they appear in the figures also  discussing appropriately the results from statistical analysis. Furthermore, The data from PLA analysis are quickly reported but the significant increase of the heteromeric complexes observed in the pathological microglia could have a role in determining the different pharmacological profile of the CB2-OX1 complex in these cells when compared to control. This aspect must be taken into consideration and discussed.

Discussion

Lines 595-596  The mechanisms for producing neuroprotective microglia, also known as  M2, are not known, although” It is not clear what the authors mean, please rephrase.

Most of the discussion section is dedicated to review the data in the literature and only a limited part aims at   discussing the results form the study. This section needs a deep revision and must be dedicated to the results of the study.

The concentration of the ligands used in the different experiments must be indicated throughout the  text. Please also pay attention to the concentration reported for each ligand in the text. For instance, caption of Figure 2:  line 447 indicates 100 μM OXA  while line 456 indicates 100 nM OXA.

The section suffers from several mistakes and deserves a deep revision for English language. See for instance:

line 401: As CB2R couples 401 to Gi protein, thus leading …… delete As

lines 404-405:….. forskolin to activate adenylate cyclase and facilitate the detection of cAMP levels decrease  …please modify the text since by activating the adenylate cyclase forskolin increases the production of cAMP but do not facilitates its detection..

line 405: In those cells: not clear the cells you refer to, please rephrase.

Line 413, control the commas

Lines 421 to 423 please control and rephrase.

Line 540 : remove that

Lines 476-479: please control the use of commas and rephrase

Lines 540-542: please control the statements and rephrase the sentences.

Please also control the discussion section.

Author Response

The   working hypothesis deserves attention since it could provide new insights of the pathological cellular events underlying the development of the neurodegenerative disorder, also suggesting new therapeutic targets for its management. Nonetheless, despite the relevance of the results, a deep revision of the manuscript is required to improve its quality and to remove the several speculations or misinterpretations of the data that reduce the relevance of the study.

The introduction is exhaustive and gives a wide overview of the available knowhow concerning the two receptors (OX1 and CB2 receptors) in the CNS also providing few information concerning their role in the Alzheimer’s disease (AD).

The methods are sufficiently described. As to the use of animals at the topo of paragraph 2.5 the authors state that “By the current legislation, obtaining protocol approval is not needed if  animals are sacrificed to obtain a specific tissue”. To the best of my knowledge however the protocol approval is required for transgenic animals as indeed reported   in paragraph 2.6.  Please clarify this point in the section 2.6 or, alternatively, remove the above reported sentence in section 2.5.

 Thank you for the comment. The reported sentence has been eliminated.

Results section

Lines 430-432: not clear why it is expected that JWH133 should not trigger any variation of calcium ion levels, taking into consideration that the CB2 receptor is coupled to the adenylil cyclase which produces cAMP taht in turn reverberate on VOCCs controlling teh influx of calcium ions within cells. Please explain or modify the text.

Thank you the comment. The reason is that calcium release is measured few seconds after ligand addition and thus, only Gq coupling receptors can induce this mobilization. The signaling pathway described by the reviewer would be detected for sure afterwards.  

Line 408: the sentence  “Ox1R can couple to different G proteins; hence we first performed assays assuming that it could couple to Gi protein.”   The statement is redundant since the authors did not discuss any other possibility despite the puzzling results observed in microglia cells from transgenic mice.

Thank you for the comment. The text has been modified according to the reviewer comment. 

Lines 473-475: The HIV-1 protein Tat efficiently permeates membranes and can enter cells and when linked to chemical it favors their internalization within cells. However the protein does not favour the viral internalization, rather it controls the efficiency of the viral transcription. Please modify the statement or remove it.

Thank you for the comment. The text has been modified according to the reviewer comment. 

Lines 533 to 535: the term small is inappropriate unnecessary and must be removed. The authors did not carried out comparative studies with the other experimental control conditions.

Thank you for the comment. The word small has been eliminated from the text. 

Line 539: again, the term counteracted is inappropriate and speculative and suggest a conclusion that is not supported by the results. Looking at the figure it emerge that when concomitantly added the two agonists cause a reduction that does not differ from that elicited by the agonist alone as well as from the FK alone. The sentence “such a negative cross-talk  that was not observed in control microglia.” is not supported by the data and sounds speculative.

Thank you for the comment. The word counteracted has been substituted by a non-additive effect. 

Lines 533- 544 The results that would support the negative (?) cross-talk linking the two receptors are confusing and over-speculative. The sentence must be modified to describe the results as they appear in the figures also  discussing appropriately the results from statistical analysis. Furthermore, The data from PLA analysis are quickly reported but the significant increase of the heteromeric complexes observed in the pathological microglia could have a role in determining the different pharmacological profile of the CB2-OX1 complex in these cells when compared to control. This aspect must be taken into consideration and discussed.

Thank you for the comment. The PLA analysis discussion has been improved.

Discussion

Lines 595-596  “The mechanisms for producing neuroprotective microglia, also known as  M2, are not known, although” It is not clear what the authors mean, please rephrase.

Thank you for the comment. The text has been improved.

Most of the discussion section is dedicated to review the data in the literature and only a limited part aims at   discussing the results form the study. This section needs a deep revision and must be dedicated to the results of the study.

 Thank you for the comment. The text has been improved.

The concentration of the ligands used in the different experiments must be indicated throughout the  text. Please also pay attention to the concentration reported for each ligand in the text. For instance, caption of Figure 2:  line 447 indicates 100 μM OXA  while line 456 indicates 100 nM OXA.

 Thank you for the comment. The concentrations have been included.

The section suffers from several mistakes and deserves a deep revision for English language. See for instance:

line 401: As CB2R couples 401 to Gi protein, thus leading …… delete As

lines 404-405:….. forskolin to activate adenylate cyclase and facilitate the detection of cAMP levels decrease  …please modify the text since by activating the adenylate cyclase forskolin increases the production of cAMP but do not facilitates its detection..

line 405: In those cells: not clear the cells you refer to, please rephrase.

Line 413, control the commas

Lines 421 to 423 please control and rephrase.

Line 540 : remove that

Lines 476-479: please control the use of commas and rephrase

Lines 540-542: please control the statements and rephrase the sentences.

Please also control the discussion section.

Thank you for the comments. The text has been improved.

Reviewer 2 Report

Major:

1. Please explain in details how you came to conclusion of tetramers formation, not dimers or mixture of different oligomers by CB2R and Ox1R. Show your calculations as necessary.

2. If your know that only TM4 and TM5 are involved in contact in a heteromer, is it consistent with a model of the tetramer - there enough structures of GPCRs to infer this information, although it is understood that the structure of class A GPCR heteromer is not available. Probably, because this is a dynamic equilibrium of various oligomeric species.

3. The legend for Figure 2 is confusing and explanation of (C-E) has to be clarified. What is shown in 2C is nowhere explained. Why fluorescence signal is negative (orange line)? What is shown in 2F is nowhere explained. Are changes for the last 5 bars (to the right) in 2F significant?

Minor:

1. Line 384: "... specific interaction between the receptors and the formation of CB2R-Ox1R heteromers with the following parameters: BRETmax = 127 ± 7 mBU and BRET50 = 110 ± 20 mB." These parameters characterize the saturation curve, not the heteromers. Please explain what exactly listed BRET parameters mean for heteromer.

Author Response

  1. Please explain in details how you came to conclusion of tetramers formation, not dimers or mixture of different oligomers by CB2R and Ox1R. Show your calculations as necessary.

Answer: Thanks for the comment. In Figure 3B, a resonance energy transfer experiment has been performed with double complementation demonstrating the possible formation of CB2-OX1 receptor complexes in transfected HEK-293T cells. It is a powerful/precise experiment in which if a saturation curve is obtained the specificity of the interaction is demonstrated (in such a heterologous system). Surely, it cannot be ruled out that this complex is part of larger oligomers, but the tetramer would be the smallest structural unit. We show that a tetrameric arrangement is geometrically feasible in a new supplementary figure (Supplementary Figure 1) with a model of the tetramer.

  1. If your know that only TM4 and TM5 are involved in contact in a heteromer, is it consistent with a model of the tetramer - there enough structures of GPCRs to infer this information, although it is understood that the structure of class A GPCR heteromer is not available. Probably, because this is a dynamic equilibrium of various oligomeric species.

Answer: Thanks for the comment. The CB2-OX1 receptor heterodimer interface is formed by a symmetric arrangement of TMs 4 and 5. However, in the same complex, there are more interfaces used, those involved in the CB2-CB2 receptor homodimer and in the OX1-OX1 receptor homodimer. We have added a new figure (Figure 5) with molecular models of the heteromer as a framework to explain a possible functional mechanism, and a supplementary figure (Supplementary Figure 1) that explores a possible tetrameric arrangement compatible with the heteromeric interface.

  1. The legend for Figure 2 is confusing and explanation of (C-E) has to be clarified. What is shown in 2C is nowhere explained. Why fluorescence signal is negative (orange line)? What is shown in 2F is nowhere explained. Are changes for the last 5 bars (to the right) in 2F significant?

Answer: Thanks for the comment. The Figure 2 legend has been modified as suggested. The decrease when cells are exposed to the OX1R antagonist (orange line) may be explained by a decrease in basal levels. It does not mean negative fluorescence, it means a lower value compared to basal line taken as reference. The changes in the last 5 bars are not significant.

Minor:

  1. Line 384: "... specific interaction between the receptors and the formation of CB2R-Ox1R heteromers with the following parameters: BRETmax = 127 ± 7 mBU and BRET50 = 110 ± 20 mB." These parameters characterize the saturation curve, not the heteromers. Please explain what exactly listed BRET parameters mean for heteromer.

Answer: Thanks for the comment. The meaning of BRETmax and BRET50 has been included in the revised version.

Reviewer 3 Report

The work entitled “Antagonization of Ox1 receptor potentiates CB2 receptor function in microglia from APPSw/Ind mice model”    by Raïch et al is a well designed and developed pharmacological work unveiling the interrelationship  among Orexin and cannabinoids receptors and fits to Special Issue: New Insight into Cannabinoid Effects 2.0

The works is developed in an elegant way, following a strategy where the authors are very comfortable. The use of primary cultures from health and APP mice is very interesting to link the possibility to use in the AD therapy by controlling microglia polarization to M2 phenotype as a neuroprotective.

The authors demonstrated the heteromeric composition for both receptors and the relation with Gi and not Gs proteins, the participation of Gq as a transduction mechanism is discussed but not clearly demonstrated (albeit an indirect measure of this activation was). The discussion part where authors try to explain the co-occurrence of two different G proteins in the tetramer is confuse. (Sentence 633-349. This paragraph should be improved to clarify the hypothesis presented by authors.

Minor spelling errors has been detected, then a carefully second looks have to be done paying attention to this details (line 74 en n.Ariched instead of enriched or line 524 Forkolin  instead of forskolin.

Author Response

The work entitled “Antagonization of Ox1 receptor potentiates CB2 receptor function in microglia from APPSw/Ind mice model” by Raïch et al is a well designed and developed pharmacological work unveiling the interrelationship  among Orexin and cannabinoids receptors and fits to Special Issue: New Insight into Cannabinoid Effects 2.0

The works is developed in an elegant way, following a strategy where the authors are very comfortable. The use of primary cultures from health and APP mice is very interesting to link the possibility to use in the AD therapy by controlling microglia polarization to M2 phenotype as a neuroprotective.

The authors demonstrated the heteromeric composition for both receptors and the relation with Gi and not Gs proteins, the participation of Gq as a transduction mechanism is discussed but not clearly demonstrated (albeit an indirect measure of this activation was). The discussion part where authors try to explain the co-occurrence of two different G proteins in the tetramer is confuse. (Sentence 633-349. This paragraph should be improved to clarify the hypothesis presented by authors.

Answer: Thanks for the comment. A new experiment whose results are displayed in Figure 3C has shown that, coupled to the CB2R-OX1R heterotetramer, there must be at least one Gi and one Gq protein. In fact, in HEK-293T cells expressing the calcium sensor and the two receptors, a Gq inhibitor and, also PTX, decreased the calcium signal induced by orexin A. These new data and the ad hoc interpretation have been included in the revised version of the paper.

Minor spelling errors has been detected, then a carefully second looks have to be done paying attention to this details (line 74 en n.Ariched instead of enriched or line 524 Forkolin  instead of forskolin.

Answer: Thanks. Errors have been corrected.

Round 2

Reviewer 1 Report

my previous concerns have been all addressed and teh manuscript is acceptable for pubblication